# The neglected giants: Uncovering the prevalence and functional groups of huge proteins in proteomes

Anibal S. Amaral[1], Damien P. Devos[1,2]*

1 Centro Andaluz de Biología del Desarrollo, Consejo Superior de Investigaciones Científicas, Campus Universidad Pablo de Olavide, Seville, Spain, 2 CNRS, Inserm, CHU Lille, Institut Pasteur de Lille, U1019-UMR 9017-CIIL-Centre d'Infection et d'Immunité de Lille, University of Lille, Lille, France

* damienpdevos@gmail.com

**Data Availability Statement:** All relevant datasets derived from the original Uniprot release are available as Supporting Information. The code used to perform all the analysis is available on GitHub at

## Abstract

An often-overlooked aspect of biology is formed by the outliers of the protein length distribution, specifically those proteins with more than 5000 amino acids, which we refer to as huge proteins (HPs). By examining UniprotKB, we discovered more than 41 000 HPs throughout the tree of life, with the majority found in eukaryotes. Notably, the phyla with the highest propensity for HPs are Apicomplexa and Fornicata. Moreover, we observed that certain bacteria, such as *Elusimicrobiota* or *Planctomycetota*, have a higher tendency for encoding HPs, even more than the average eukaryote. To investigate if these macro-polypeptides represent "real" proteins, we explored several indirect metrics. Additionally, orthology analyses reveals thousands of clusters of homologous sequences of HPs, revealing functional groups related to key cellular processes such as cytoskeleton organization and functioning as chaperones or as E3-ubiquitin ligases in eukaryotes. In the case of bacteria, the major clusters have functions related to non-ribosomomal peptide synthesis/polyketide synthesis, followed by pathogen-host attachment or recognition surface proteins. Further exploration of the annotations for each HPs supported the previously identified functional groups. These findings underscore the need for further investigation of the cellular and ecological roles of these HPs and their potential impact on biology and biotechnology.

## Author summary

In our study we focused on a fascinating and often overlooked aspect of biology: proteins that stand out due to their extraordinary length, with more than 5000 amino acids in size. These exceptional proteins, aptly named "huge proteins", have been found in surprising abundance across various life forms, with over 41 000 identified. Most of these HPs belong to eukaryotes. Within this group, specific branches like Apicomplexa and Fornicata exhibit a remarkable tendency to encode for these HPs. Astonishingly, certain types of bacteria, like *Elusimicrobiota* and *Planctomycetota*, encode more of these HPs than the average eukaryote. To confirm the significance of these findings, we used various methods and discovered that these mega-molecules are indeed authentic proteins with crucial roles

https://github.com/joaosegurilho/huge_proteins_2023.

**Funding:** DPD was funded by the Gordon and Betty Moore Foundation grant #9733. The funders had no role in study design, data collection and analysis, decision to publish, or preparation of the manuscript.

**Competing interests:** The authors have declared that no competing interests exist.

in cellular activities. In eukaryotes, they participate in vital processes such as maintaining the cell's structure and acting as cellular caretakers. For bacteria, these proteins are involved in intricate tasks like producing essential compounds and interacting with host organisms. This study sheds light on these enigmatic biological giants, urging further exploration into their roles, opening new avenues in the realms of biology and biotechnology.

## Introduction

The majority of proteins have a length between 100 and 500 aminoacids (aa) [1]. The distribution of these lengths has been found to adhere closely to a log-normal or to a gamma function [1]. While small peptides, have been extensively studied elsewhere [2,3], the investigation of significantly large proteins has received limited attention thus far. Our observations reveal the presence of numerous proteins exceeding 5000 aa in length, which we termed "Huge Proteins" (HPs).

In a recent survey of 2326 species, the authors demonstrated that protein length distributions are consistent across species [1]. They also reveal that some outliers, more often than not, small proteins, were artifacts when comparing proteomes from closely related species. Nevertheless, some of these extremely large outliers are well characterized and known to exist. Possibly the most famous such outlier is the filamentous protein Titin, the most abundant muscle protein after myosin and actin in Vertebrates. This protein forms a myofilament system in skeletal and cardiac muscles [4,5]. HPs have also been detected in model organisms, as is the case for the *Drosophila melanogaster* Futsch protein [6], the *Caenorhabditis elegans* Mesocentin [7], the *Arabidopsis thaliana* Midasin [8] or the *Danio rerio* Mysterin [9]. However, their roles in the cell have not thoroughlybeen investigated.

Previous analyses reported 3732 genes longer than 5kb in a limited dataset of 580 prokaryotic genomes [10]. They found that giant genes are strain-specific, differ in their tetranucleotide usage from the bulk genome, occur preferentially in non-pathogenic environmental bacteria and are thought to be expressed only in conditions where the organism is not proliferating [10]. These huge genes coding for HPs have been categorized in two main functional groups: Surface proteins and Non-Ribosomal Peptide Synthetases (NRPS)/Polyketide Synthetases (PKS). NRPS are large enzymatic complexes responsible for synthesizing microbial secondary metabolites called non-ribosomal peptides that show high structural diversity and perform a multitude of functions. Working typically in sequential order resembling an assembly line, these synthetases incorporate an aminoacid and pass it along the multi-domain protein or several proteins to build a final peptide [11]. An alternative to this system is the PKS, also reliant on the same core mode of action, but producing polyketides instead of peptides. Contributing to the large complexity of final products that these systems are able to produce, some genes were found to encode hybrid NRPS-PKSs resulting in the synthesis of polyketyde-peptide hybrids [11]. On the other hand, one example of a surface protein is the extracellular matrix-binding protein homologue from *Staphylococcus aureus*. This protein, with almost 10 000 aa, has been found to bind fibronectin produced by the host (in this case Human) [12]. More and more, HPs have been reported from the characterization of particular strains, including in *Fuerstia marisgermanicae* [13], in four Isosphaeraceae planctomycetotales [14], and the recent descriptions in Omnitrophota species of many HPs including some of the largest ever reported, exceeding the size of eukaryotic HPs [15]. However, recent broad analyses, pertaining to the existence and distribution of these proteins, are missing.

In order to conduct this analysis, we decided to address a broad range of questions regarding recently observed very long protein sequences. In order to sustain the rest of the study, we tried to assert whether these are real sequences or erroneous predictions. We further questioned if these sequences were restricted to a particular set of organisms. Finally, based on homology, domains or structural features we tried to group them into functional categories. With this work we set out to survey HPs diversity and function.

## Results

### Where do they occur?

We report over 41 754 proteins of size $>= 5000$ aminoacids in UniprotKB (Fig 1A). As expected the majority of HPs comes from eukaryotic proteomes (Fig 1B), with a sizable proportion coming from bacteria and relatively few from archaea.

There seems to be a tenuous relation between proteome size and the number of HPs encoded (Fig 1C). The proteomes with the highest number of HPs belong to two Echinodermata species: *Patiria miniata* (Bat star) with 250, followed by the sea urchin *Strongylocentrotus purpuratus* with 234. The biggest proteome in our dataset from *Araneus ventricosus* contains around 246 000 proteins, with 14 mostly non-homologues HPs. In the case of bacteria, the biggest proteome is from *Streptomyces* sp. SID7982 with approximately 19 000 proteins, from which only 1 is a HP. The bacterial strains with the highest number of HPs are *Streptomyces rapamycinicus* NRRL 5491 and the Desulfobacteria *Candidatus* Magnetomorum sp. HK-1 with 31 and 30, respectively. Again, both sets were mostly composed of non-homologue proteins. Finally for Archaea, the largest proteome from the uncultured *Cand. Pacearchaeota* archaeon contain only 1 HP, while the Euryarchaeota strain *Methanobrevibacter* sp. YE315 contains 7 HPs.

In terms of likelihood of a specific phylum to encode HPs (Fig 1D), most phyla specially the ones belonging to either archaea or bacteria, have a median likelihood of 0% or very close, which is only the case for Microsporidia, in eukaryotes. On the opposite end, phyla like Apicomplexa, Fornicata, Cnidaria, Euglenozoa have a considerable median likelihood ranging from 0.11 to 0.40% (*i.e.* one to four HPs out of one thousand proteins). Chordata show a median likelihood of 0.04%, in the expected range for eukaryotes. Remarkably, some bacterial phyla are enriched in HPs, in particular, *Cand.* Peregrinibacteria, Elusimicrobiota, *Cand.* Omnitrophica, *Cand.* Margulisbacteria, planctomycetota, and Lentisphaerae (Fig 1D). These have between 0.03% and 0.08% likelihood of containing HPs, comparable to the average for eukaryotic phyla. Four of these six bacteria with most HPs belong to the Planctomycetota-Verrucomicrobiota-Chlamydiota (PVC) superphylum, a taxonomic group previously highlighted for its peculiar biology [16,17].

In conclusion, there is a nontrivial number of HPs in a significant number of genomes. This is particularly evident in eukarya, but some bacterial phyla also encode a high proportion of HPs.

### Do they exist?

The mere existence of such HPs is surprising. This is due to the fact that the probability of encountering a stop codon increases exponentially with the length of a sequence, while the chance of accumulating deleterious mutations increases. We thus interrogated the evidences supporting the reality of their existence. We could detect statistically significant differences (Kruskal-Wallis and One-way ANOVA with 1.51e-171 and 1.20e-118, respectively) of annotated evidences between the trends of HPs and Non-HPs (Fig 2A). This difference in trends is related with higher percentages of HPs with "Evidence at protein level", when compared to

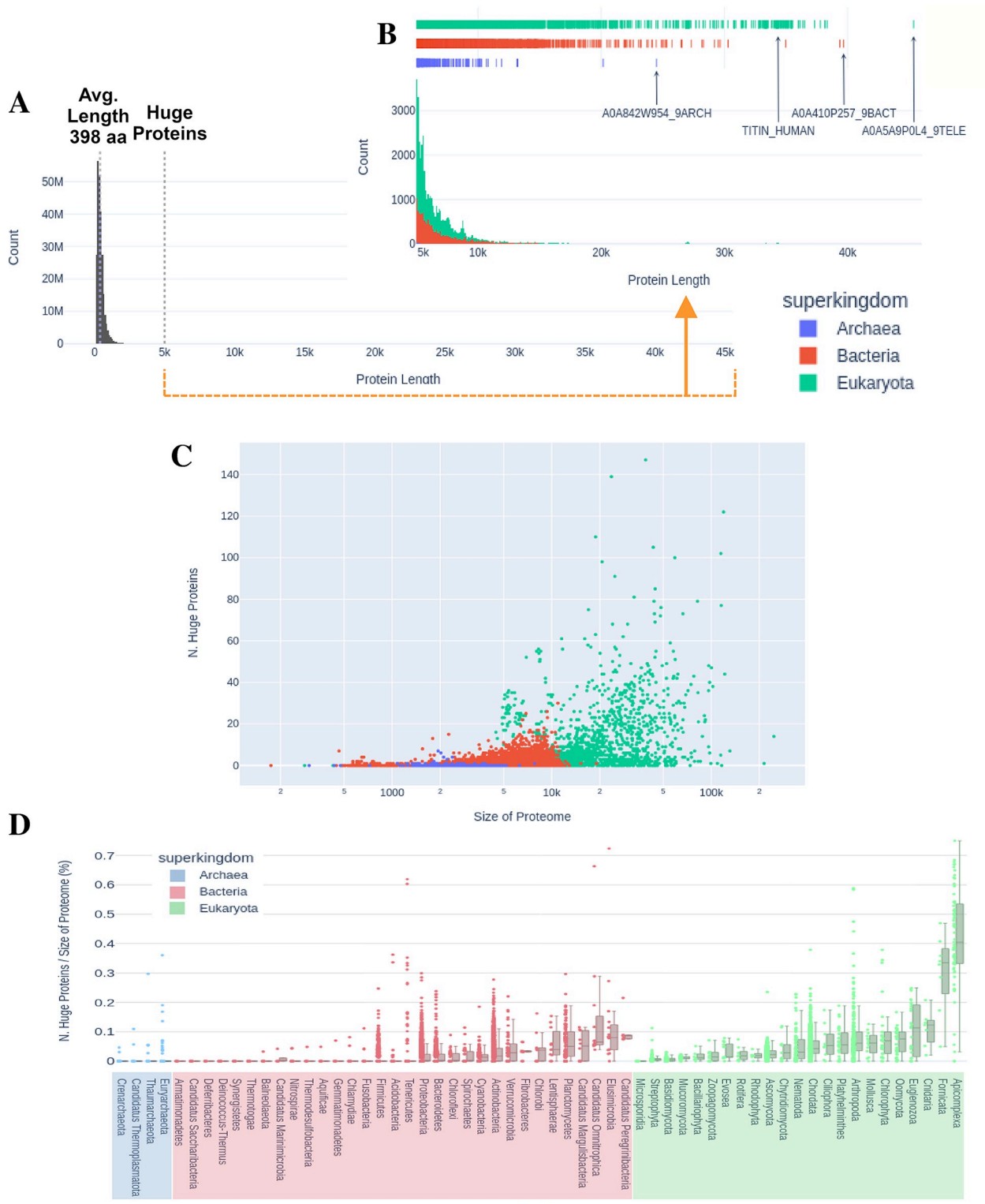

**Fig 1. Protein length distribution across the three superkingdoms of life.** (A) Whole distribution for all proteins in Uniprot (Release 2023_01). (B) Inset of the Distribution, for protein lengths bigger than 5000 amino acids, by superkingdom. Relevant proteins are highlighted. (C) Plot of the relationship between the size of the proteome (number of proteins), at log scale, and the number of huge proteins. (D) Box plot showing the likelihood of a proteome to contain huge proteins, per phylum, in percentage (number of huge proteins divided by proteome size). Each dot is a single proteome. All plots but A are colored blue, red and green for archaea, bacteria and eukaryota, respectively.

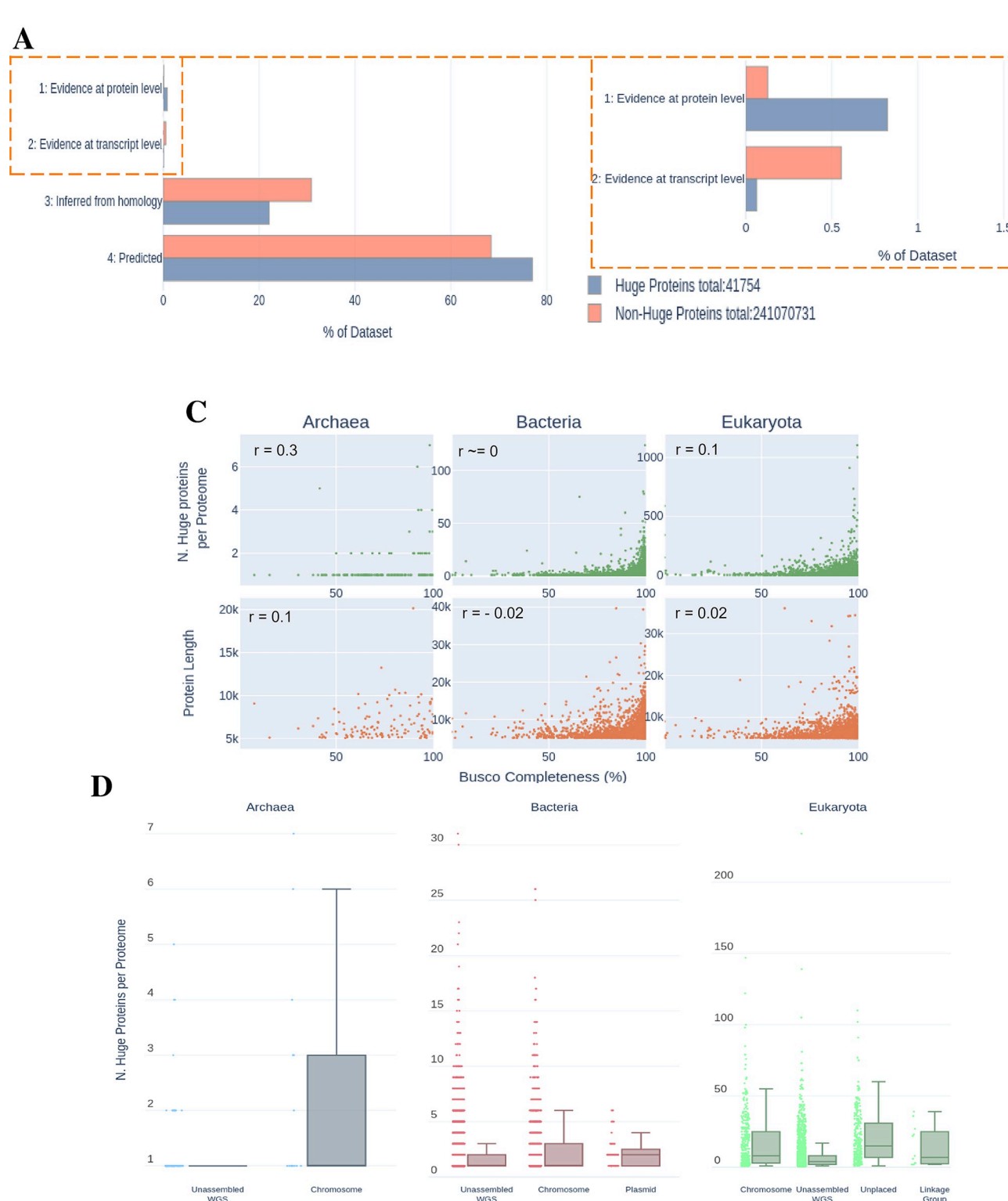

**Fig 2. Likelihood of Huge proteins Existence.** (A) Distribution of the proportion of the dataset for each Protein Existence level (Uniprot), either for Huge proteins (Blue) or Non-Huge proteins (Orange). (B) Inset of A, zoom of the first two levels. (C) Scatter plot comparing the number of huge proteins per proteome (top row, green, each green dot is a proteome) or Protein Length (bottom row, red, each dot is a protein) with the BUSCO Completeness score for each proteome. For archaea, bacteria and eukaryota, from left to right, respectively. (D) Box plot relating the Number of Huge proteins per Proteome with the Assembly level of the genomes. For archaea, bacteria and eukaryota, from left to right, respectively. Colored by Assembly level category.

Non-HPs. There is no strong correlation between the number of HPs and the completeness of the genome (Fig 2C). If anything, a slight positive correlation between genome completeness and number of HPs can be appreciated. The same trend holds true when comparing Protein length with genome completeness. Similarly, no correlation between the genome assembly levels and the number of HPs per proteome could be detected (Fig 2C).

Concerning Uniprot Existence levels, the overall trend is equal when comparing HPs and their counterparts (Fig 2A). Around 77% of HPs are reported as "Predicted", with only 22% approximately being "Inferred from Homology". Although in a higher proportion than Non-HPs, only 0.8% of HPs have "Evidence at protein level". Fewer still (0.06%) are assigned with "Evidence at transcript level". (Fig 2B).

Regarding the relation between BUSCO Completeness Score and either Number of HPs or Protein length, we observe an accumulation closer to 100% completeness, respectively.

When compared to the Genome Assembly level (Fig 2C), a higher number of HPs seems to be linked to proteomes derived from higher quality genomes, in the case of bacteria and archaea. Although most proteomes are derived from "Unassembled WGS", a significant proportion comes from "Chromosome" level assemblies, with both having the same median number of HPs per proteome. Remarkably, 67 bacterial plasmids have at least one HP. Archaea have at most 7 HPs in one single proteome, with the majority of these present in proteomes assigned as "Chromosome" level assemblies. When analyzing eukaryotes a more complex picture arises, with many types of assemblies. The majority of proteomes belong to "Unassembled WGS", "Chromosome" or "Unplaced" (i.e. the sequence is not associated with any specific chromosome, and as such, with unknown location and orientation). Here the highest median number of HPs are present in "Chromosome" and "Unplaced", with the median number for "Unassembled WGS" being considerably lower. The assembly level with the lowest median value is "Linkage Group" that also has the lowest number of observations.

In conclusion, HPs follow the same trend in terms of existence levels as Non-HPs, with almost 1% annotated as "Evidence at protein level". Furthermore, there seems to be a predominance of HPs in higher quality genomes. A slight correlation between the number of HPs or Protein length and genome completeness was detected. Thus, we observe a correlation between the number of HPs and genome quality.

## Clusters of orthologues

One important argument in favor of the existence of HPs is that multiple copies of similar proteins exist in different or the same genome. We thus clustered HPs according to sequence similarity, and observed that HPs do indeed exist in multiple copies, further supporting their existence as "real" proteins. We observed clusters of similar proteins existing in different organisms, some of which can be phylogenetically far apart. At the same time we performed Cluster of Orthologues (COG) assignment in order to functionally characterize each cluster. using non-huge orthologues.

Some clusters are phyla-specific, while others are taxonomically diverse (Fig 3A–3C). This observation holds true independently of the superkingdom. Meaning that some HPs are innovations of specific groups. Others are broadly conserved HPs, suggesting that they were retained throughout evolution or are products of horizontal-gene transfer events.

Out of a total of 7183, only 2492 clusters have two or more proteins (Fig 3). The rest are all singletons given the strict parameters used (minimal sequence identity of 0.3 and alignment coverage of 0.8 in mmseq2). Very few clusters had proteins from different superkingdoms, all being predominantly from bacteria, classified as NRPS, with at most one or two proteins from eukaryota.

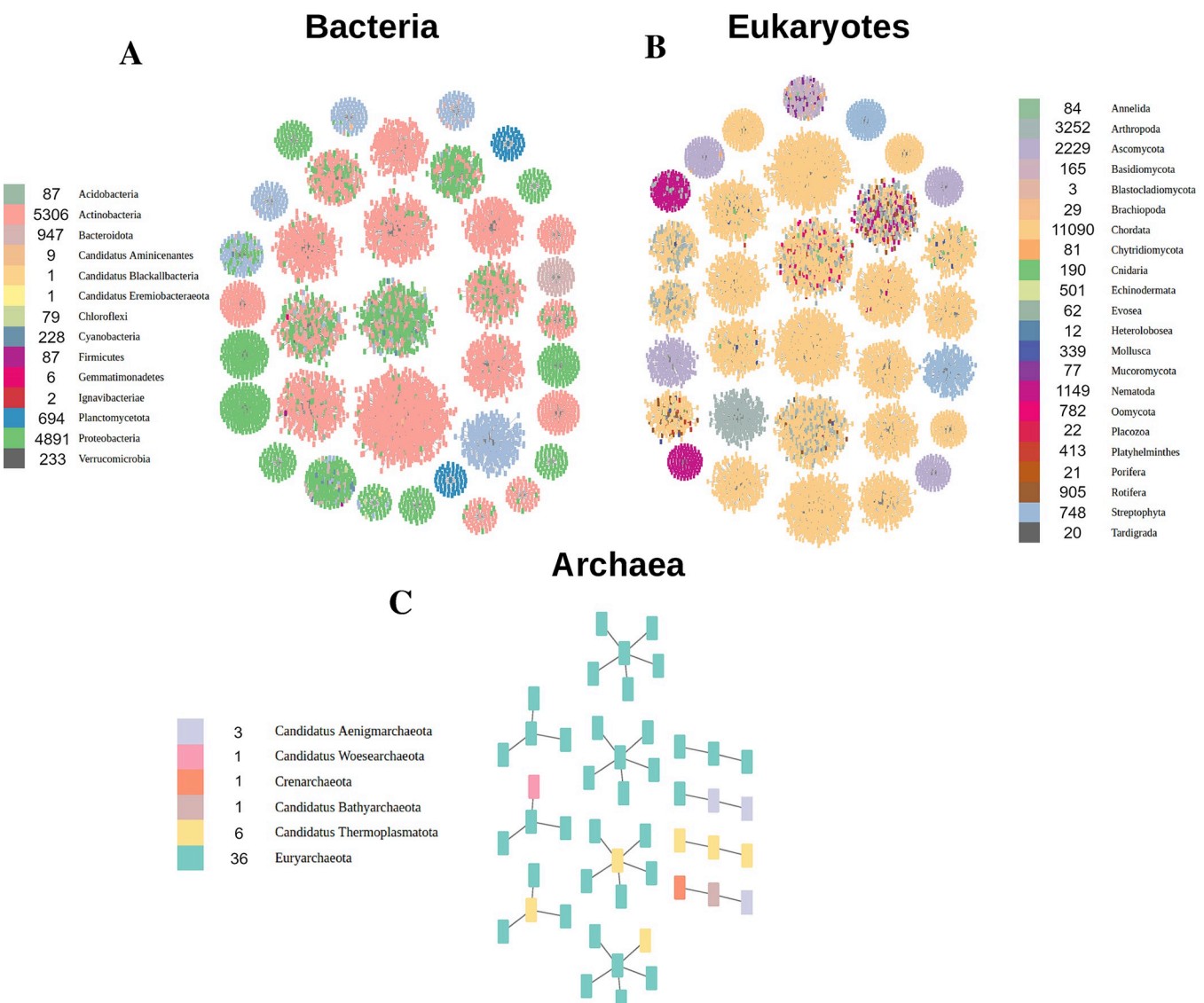

**Fig 3. Huge proteins clusters for each kingdom.** Only the largest clusters are shown for A) bacteria (top 36) and B) eukaryotes (top 32). All clusters are shown for (C) archaea. Proteins inside the clusters are represented by dots colored by phyla. Total number of HPs in dataset is shown for each phyla.

The majority of Eukaryotic clusters encompass functional categories found in a broad range of phyla (Fig 3B). Examples of such functions are: "Cytoskeleton" composed mainly of Plectin, Dystonin, Nesprin-1 and 2 and Titin; "Signal transduction mechanisms" where we find the majority of Titin and also Nebulin, Obscurin, Ryanodine receptor 1; "Transcription" is only composed of Histone-lysine N-methyltransferases and some putative Hedgehog proteins; "Post-translational modification, protein turnover, chaperones" mainly composed of E3 ubiquitin ligases and transferases of the HECT or RING types, and finally "Intracellular trafficking, secretion, and vesicular transport" that is mostly composed of Piccolo, Chorein and the nuclear chaperone Midasin. A small subset of these are putative "Extracellular structures" containing Usherin and SCO-spondin. Around 3800 have "Function Unknown", meaning that they belong to COGs but their function is undeciphered. As observed previously, the two eukaryotes with the highest likelihood of having HPs are Apicomplexa and Fornicata. In the

case of the former, HPs from this phylum share the most common eukaryotic COGs functional categories, except for "Inorganic ion transport and metabolism | Signal transduction mechanisms" and "Extracellular structures". Finally, Fornicata HPs were only assigned to "Cytoskeleton" and "Function Unknown".

Most bacterial clusters can be grouped in two main functions: by far the most abundant is "Secondary metabolites biosynthesis" that encompasses proteins such as NRPS or PKS proteins; these are followed by "Cell wall/membrane/envelope biogenesis" and "Intracellular trafficking, secretion, and vesicular transport" that mostly group Cadherin-containing or DUF11-containing proteins and Hemagglutinins, respectively. As for HPs belonging to the group of bacteria with the highest likelihood, *Cand.* Peregrinibacteria, Elusimicrobiota, *Cand.* Omnitrophica, *Cand.* Margulisbacteria, *Lentisphaerae* and *Planctomycetota*, only the proteins of the latter were assigned COG's, highlighting the poor characterization of the others.

In the case of archaea, very few proteins were assigned to COGs, all from Euryarchaeota. The most common of these functional assignments are "Replication, recombination and repair" and "Lipid transport and metabolism". For the former, all are Halomucins that may be related to protection from desiccation, rigidity, and maintenance of cell morphology. As for the latter, all 3 seem to be either a PKS or a NRPS/PKS hybrids (Table 1).

Around 1442 proteins have no homology inside our dataset, but are assigned to COGs, suggesting a possible recent fusion of smaller proteins into HPs.

In conclusion, the grouping of HPs in clusters of sequence related proteins is a strong argument in favor of their existence.

**Potential artifacts.** Thus, the vast majority of HPs appear to be correctly inferred proteins. Despite this, we were able to find indications that some could be in fact errors of gene prediction methods.

**Table 1. Most common functional assignments, sorted by superkingdom and number of proteins in cluster.**

| Superkingdom | COG category | COG description | N. Proteins |
|---|---|---|---|
| Eukaryota | Z | Cytoskeleton | 5765 |
| | T | Signal transduction mechanisms | 5228 |
| | S | Function unknown | 3808 |
| | Q | Secondary metabolites biosynthesis, transport and catabolism | 1088 |
| | O | Post-translational modification, protein turnover, chaperones | 1017 |
| | U | Intracellular trafficking, secretion, and vesicular transport | 849 |
| | K | Transcription | 710 |
| | PT | Inorganic ion transport and metabolism | Signal transduction mechanisms | 481 |
| | W | Extracellular structures | 330 |
| Bacteria | Q | Secondary metabolites biosynthesis, transport and catabolism | 7962 |
| | M | Cell wall/membrane/envelope biogenesis | 965 |
| | U | Intracellular trafficking, secretion, and vesicular transport | 381 |
| | IQ | Lipid transport and metabolism | Secondary metabolites biosynthesis, transport and catabolism | 262 |
| | D | Cell cycle control, cell division, chromosome partitioning | 143 |
| | S | Function unknown | 131 |
| | G | Carbohydrate transport and metabolism | 99 |
| | N | Cell motility | 92 |
| | QU | Secondary metabolites biosynthesis, transport and catabolism | Intracellular trafficking, secretion, and vesicular transport | 85 |
| Archaea | L | Replication, recombination and repair | 6 |
| | I | Lipid transport and metabolism | 3 |

There are around 3500 proteins with no PFAM domain prediction. Also, approximately 3200 proteins have no homology inside our dataset and no COGs assignment. Finally, there are 33 proteins from eukaryota and bacteria that have at least 99% disorder prediction, with 15 of these having exactly 100% disorder. These proteins can thus be artifacts of prediction methods, although there is the possibility of being entirely disordered novel proteins. (S1 and S2 Tables).

## What is their function?

Having explored the evidence in favor of the existence of such HPs, we now inquire about their possible function. We explored this from two complementary perspectives: domain assignments/architecture and protein features and subcellular localization.

**Protein domains and architectures.** From the initial 41 754 HPs, only 38 248 were assigned, through hmmscan targeting PfamA, with at least one PFAM domain. Still, the majority of proteins were annotated, allowing for better understanding of their putative functions.

In the case of bacteria (Fig 4A), just from the straightforward assessment of the distribution of single domains, the grouping of functions appears distinctively. For instance, the first four positions in the distribution, PP-binding, AMP-binding, AMP_binding_C and Condensation domains all have a similar taxonomic profile, with the biggest contribution coming from Actinobacteria. This is due to the fact that all these domains are integral parts of NRPS proteins (Fig 4D). This is further reinforced by the presence of several groupings along the distribution that are related to either NRPS or PKS systems. Positions 11, 12, 15, 18 and 21 have a different taxonomic profile corresponding to Big_9, Cadherin_5, Cadherin_4, HemolysinCabind and Dockerin_1, respectively. These are observed in extracellular proteins with a diverse range of functions. The biggest contribution for this last group is from *Planctomycetota*. Finally, DUF_11 (position 23) stands out as having a completely unrelated taxonomic profile. Interestingly, Dockerin_1 domain is a canonical part of the cellulosome, an extracellular protein complex that has been extensively studied in Firmicutes, but relatively poorly characterized in other bacteria phyla. We report that it is found in higher proportion in HPs from *Planctomycetota*. In terms of protein architecture, the most frequent for bacteria are NRPS followed by PKS proteins (Fig 4D).

For eukaryotes, Spectrin domain followed by members of Calponin homology (CH) and IG Immunoglobulin (IG) clans dominate the HP landscape. These domains are present in a broad range of proteins with varyious functions. These groups are dominated by Chordata, Nematoda and Arthropoda. Another distinctive category, in terms of taxonomy, groups together domains from the P-loop_NTPase clan that often perform chaperone-like functions. The most common HP architecture in eukaryotes is E3-ubiquitin ligases, that functions as recognition and facilitators of ubiquitination for downstream processing of other proteins. The second most common architecture is a Histone-lysine N-methyltransferase.

Finally, for archaea, the most common domain, by a large margin, is Laminin_G_3. This domain is part of the broad Concanavalin clan, that includes a diverse range of carbohydrate binding domains and glycosyl hydrolase enzymes. This domain is present in almost all archaeal HPs. Next, the Big_3_5 (Bacterial Ig-like domain group 3) domain mostly present in Euryarchaeota. Interestingly, two domains SGL (SMP-30/Gluconolactonase/LRE-like region) and Lactonase are exclusive to Thaumarchaeota. The most common architecture in archaea, contains one copy of Laminin_G_3 found at the N-terminal. Some variations of this architecture were found, with two or more of the same domain, around the same region. The function of these proteins is not understood. The second most common architecture is reported as an "Uncharacterized protein" mostly composed of CarboxypepD_reg, Big (13 or 3_3) and Glucodextran_B.

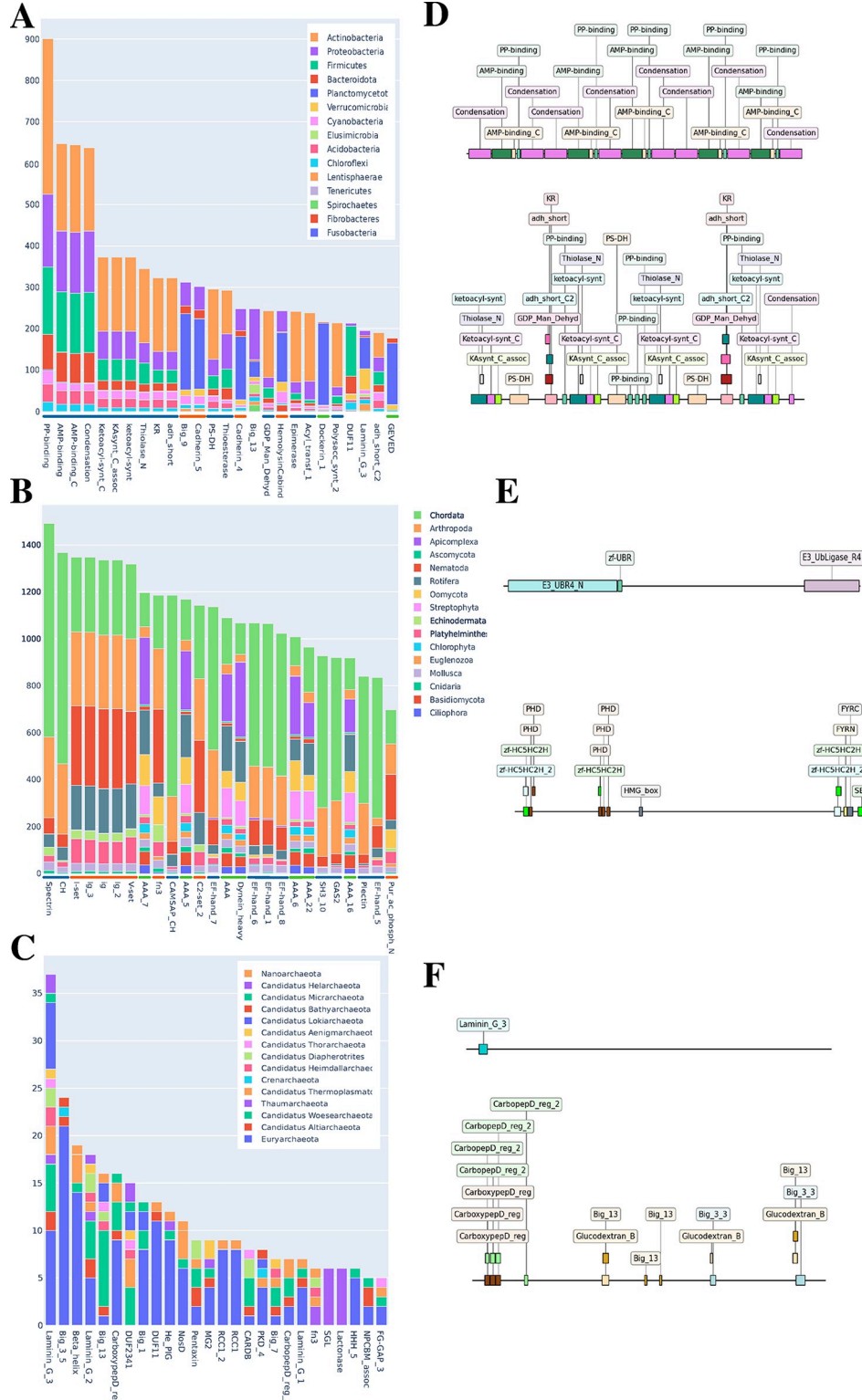

**Fig 4. Domain composition and architecture of Huge proteins.** (A-C) Stacked bar distribution of PFAM domains, from A to C is bacteria, eukaryota and archaea, respectively. Each bar in the distribution is colored based on contribution from each phyla. Calculations were performed on a restricted subset limiting the maximum number of proteomes from each phylum to 50, as to not skew the distribution. Only the top 15 phyla in terms of contribution is shown, except for archaea. The distinct taxonomic pattern of each bar illustrates the differing contributions from each

phyla. (D-F) Most common architectures of bacteria, eukaryotes and archaea, respectively. Sizes of the proteins are representative and not to scale.

**Protein features and localization.** We first observed that bacteria and eukaryota have highly variable disorder percentages. On the contrary, archeal HPs accumulate closer to 0%, with very few passing the 50% mark. Eukaryotes and bacteria HPs have a higher tendency for disordered regions, with a slight difference in distribution when compared to Non-HPs, which tends to accumulate closer to 0%.

The presence of signal peptide is almost limited to HPs shorter than 20 000 aminoacids, with proteins above this size being unlikely to have a signal peptide (S3B Fig). There is no clear trend in terms of HPs having transmembrane helices (TMHs) (S3C Fig). Most HPs have no TMH, but many have at least 1 TMH.

An intriguing observation reveals narrowing of annotation values at around the 26k to 35k protein length range (S3D and S3E Fig). HPs within this range rarely contain signal peptides and very few possess transmembrane helices. Considering only eukaryotic proteins, these tend to have between 10% to 30% disorder, which is in agreement with Non-HPs.

Furthermore, in terms of PFAM domains termed as "Repeats" we found exactly 15400 proteins with at least one of such domains, out of a total of 38248 with domain assigned. This result hints at a possible evolutionary trajectory of these sequences, through the accumulation of repetitive motifs.

Analysis of Gene Ontology (GO) terms (S4 Fig) reveals that most bacterial HPs are located at the "cell periphery" or "membrane", with a significant number either at an "intracellular anatomical structure" or at "external encapsulating structure". A contrasting picture is seen with eukaryotes, where most proteins are found at "intracellular anatomical structure", "organelle" or "membrane".

As for the possible involvement in a Biological Processes, the majority in eukaryotes are related to "cellular component organization or biogenesis" and approximately 2500 are involved in "organic substance metabolic process" or "intermediate filament-based process". In the case of bacteria, most are either "organic substance metabolic process" or "biosynthetic process". Finally, for archaea "cell adhesion", "organic substance metabolic process", "primary metabolic process" or "nitrogen compound metabolic process" are the most common terms assigned.

Lastly, in terms of Molecular Function, eukaryotic HPs are mostly involved in "protein binding", "ion binding" or "small molecule binding". In bacteria, they relate to "small molecule binding", "amide binding" or "modified aminoacid binding". In the case of archaea, the vast majority was assigned the term "ion binding".

## Discussion

We found a considerable number of proteins with more than 5000 aminoacids (Fig 1). As expected, the majority belongs to eukaryotic proteomes, followed by bacteria and very few representatives from archaea. There is a tenuous relation between the number of proteins per proteome and the number of HPs found. To explore this, we calculated the quotient of these two components which revealed a tendency for some phyla to have more HPs. This tendency was most noticeable in eukaryotes. Despite this, the mechanisms behind the observation that a few bacteria phyla have a higher tendency than some eukaryotes are still to be elucidated, and possibly linked to niche specialization [19].

The higher tendency for some eukaryotic phyla to contain HPs, particularly, Apicomplexa, as been observed in other studies. Nevers and colleagues, explored the protein length

distribution across Eukaryotic and Prokaryotic proteomes [1,19]. They found that proteomes, with protein lengths distributions outside the norm for their particular phylum could hint at possible protein prediction artifacts. In the case of proteomes with "long proteins", such as Apicomplexa and the fungal genus *Ustilago*, and despite previous reports of incorrect gene prediction in Apicomplexa, they found no evidence for these proteins to be artifacts. For *Ustilago*, that belongs to the phylum Basidiomycota, there seems to be a very small likelihood to possess HPs in our data. Interestingly, besides Apicomplexa, we also observed that Fornicata had similar tendency, which hints at a possible biological connection between high numbers of HPs and the parasitic/commensal lifestyle. Contrary to eukaryotes, from the bacterial phyla with a high propensity for HPs, *Cand*. Peregrinibacteria, Elusimicrobia, *Cand*. Omnitrophica, Lentisphaerae, have been shown to have a host-associated lifestyle [20–22], suggesting that bacterial HPs could be associated with this kind of behavior. Of further note, around 23 HPs were found in the Asgard clade. Of these, 6 belonged to the phylum *Cand*. Heimdallarchaeota, 2 to *Cand*. Helarchaeota, 12 to *Cand*. Lokiarchaeota and 3 to *Cand*. Thorarchaeota. All are "Predicted" except one that is "Inferred from homology".

As mentioned previously, the existence of these proteins can be questioned *a priori* owing to their extreme length. We explored two approaches. We used indirect measures such as Uniprot's protein existence level, BUSCO genome completeness score, and genome assembly quality. From this we observed that there is a considerable percentage of HPs with "Evidence at protein level", more than for Non-HPs (Fig 2A and 2B). Of course, these two values are not directly comparable, but allow for an understanding of the trend in both cases. Besides the proportion, we tested if sampling the Non-HP dataset, where each sample size is equal to the total number of HPs, would result in a similar trend. This didn't occur, with the trend being independent of the number of samples taken. The small percentage of HPs with "Evidence at transcript level" could be due to several reasons: HPs are potentially expressed under specific conditions that are not met in expression assays; these proteins are part of the repertoire of organisms were these experiments were not performed, or due to their presence outside the core pangeome, has is known in the case of many bacteria HPs [10]; the ones that are found are better characterized and were thus assigned a higher Evidence level. We also observed an increase in HP length and in the number of HPs per proteome that correlates with higher Completeness scores, reinforcing the idea that the overall majority of HPs comes from higher quality genomes. The last measure revealed that most HPs, from bacteria and archaea, belong to proteomes derived from higher quality genomes (Fig 2D). Although most proteomes were obtained from "Unassembled WGS", "Chromosome" or "Unplaced" assemblies the majority of HPs were derived from good quality assemblies types such as "Chromosome" or "Unplaced". The fact that around 67 HPs were found in "Plasmid" level assemblies, reveals the possibility that some HPs might be a product of horizontal-gene transfers.

Based on this first approach, we were able to find in our dataset a selection of proteins that have expression data associated. These HPs were either classified has having "Evidence at transcript level" or "Evidence at protein level". From the 26 that were assigned the former, we found proteins belonging to eukaryotes and bacteria. One example is the 11 872 amino acids long Nonribosomal peptide synthetase LcsA, from an Ascomycota, that has been found to be expressed [24]. Regarding the ones with the highest level of evidence we have the case of Bltp1 from *Mus musculus* (A0A179H164) that is expressed in several tissues at different developmental stages (ExpressionAtlas: ensmusg00000037270).

In the second approach, we showed that these HPs could be clustered into orthologues groups emphasizing that they are not isolated artifacts, but biologically meaningful. The exclusivity of some clusters for a specific phyla reveals that many of these proteins are a product of intra-phylum innovations, while others are conserved in a broad range of organisms (Fig 3).

The major clusters obtained were formed by functionally related proteins, regardless of super-kingdom. Most Eukaryotic clusters were related with key cellular processes or functions such as cytoskeleton organization, chaperones or E3-ubiquitin ligases. The biotechnologically relevant PKS and NRPS systems were the biggest contributors in bacteria, followed by a broad range of extracellular/surface proteins with high functional diversity (ex: pathogen-host attachment, signal reception, transport, or adhesion). This was also reinforced by the domain groupings seen in Fig 4A, where the most common domains were related with either NRPS, PKS, or characteristic of extracellular proteins. All of this validates the functional categories of giant genes in Prokaryotes, first postulated by Reva and Tümmler [10].

The observation that around 1442 proteins had no homology inside our dataset but assigned COGs, hints at the potential uniqueness of these among HPs, only having similarity with other Non-HPs.

Despite our initial survey, there are still many mysteries and questions left to answer. Considering an evolutionary perspective, what is the origin of these proteins? The most prominent hypothesis is that they arise through degeneration of stop codons leading to fusion of neighboring genes [1,10,25]. The observation of "degenerated stop codons" in genes coding for HPs, i.e. codons resulting from non-synonymous substitutions of originally stop codons could elucidate this question. Another possibility is that these proteins are a result of repetitive expansions of codons from the same gene as is most likely the case for the E3-ubiquitin ligases found (Fig 4E), where the flexible part between the two domains expanded. A further hypothesis is the fusion of paralogues as could be the case for most NRPS and PKS here reported.

The function of these proteins is, for the most part, poorly understood. There are many reasons for this. First, the vast majority of proteins in the databases are not functionally characterized. Second, is the difficulty in deciphering their structure through traditional methods or even relying on state-of-art methods such as Alphafold2 [26]. The latter is mostly limited by the major resource requirements in terms of computational power and memory, which increase with sequence length. Another, is the lack of experimental characterization of many non-model organisms, and by consequence their proteins. Thus, the molecular and cellular behavior of these proteins needs investigation. As shown previously they are most likely expressed, although evidence of their expression could clarify their existence without depending on indirect measurements. Despite this, their fate after synthesis is still unclear. Some could be subjected to post-translational modifications. Some could be cleaved to originate smaller functional proteins. The difficulty here is in the large scale prediction of the proteases present in a specific organism, the motifs that these proteases might recognize, and if they in fact cleave HPs present in this organism. Some could be forming complexes with other proteins.

Some authors have also proposed that the synthesis of these proteins could have a significant energetic impact on the cells [1,10,27]. This claim however, lacks empirical evidence to be sustained, having only been shown theoretically. This cost could be negligible or equal to synthesizing the full operon (from where the HP might have originated).

Despite this, many eukaryotic HPs, are badly characterized. Understanding these proteins will lead to a better understanding of their physiological roles. Many of these proteins, mostly present in bacteria and fungi, are implicated in secondary metabolite production. It is the case of the NRPS and PKS, which have demonstrated the capability of generating numerous metabolites that are relevant to biotechnology, including some with pharmaceutical properties.

Reflecting on the characterization levels of these HPs, considering the full extent of annotation altogether with the previous literature on the subject emphasizes the lack of research done towards understanding the cellular and ecological roles of these remarkable proteins. Many questions, not only mechanistic, are still left unanswered.

## Methods

All analysis, data retrieval and manipulation was performed using Python 3.9, Plotly (version 5.11.0), Pandas (version 1.4.4), Numpy (version 1.22.4), Ete3 (version 3.1.2), Matplotlib (version 3.6), DnaFeaturesViewer (version 3.1.1), NetworkX (version 2.8.4), Obonet (version 0.2.3), Scipy (version 1.7.3), Cytoscape (version 3.9.1).

### Dataset

The initial dataset was downloaded from UniprotKB [28] (Release 2023_01). The database was queried for proteins bigger than 5000 amino acids, that did not belong to Viruses (NCBI: txid10239) proteomes, thus obtaining the final HP dataset. Sequences in this dataset followed the same standard of quality cut-off as UniprotKB. From this we obtained 9946 proteomes in total, with varying degrees of completeness, 7224 belonged to bacteria, 2594 to eukaryota and 128 to archaea.

### Prevalence

The full distribution of sizes was plotted using all protein "ids" and the respective protein length, for each protein in the initial dataset. The same was followed for the distribution of proteins bigger than 5000 amino acids.

To obtain the comparison between number of HPs and proteome size, we first queried UniportKB Proteomes to obtain all the proteomes (either containing or not containing HP), retrieving the respective total number of proteins in proteome (Size of Proteome) and the NCBI taxid. Lastly, we filtered the proteomes: to exclude the ones belonging to phyla with less than 5 proteomes overall; all with size of proteome smaller than the mean minus the standard deviation for the respective phyla and finally, removed all that belonged to plasmid proteomes. A scatter plot was constructed, applying the logarithmic scale to the axis that showed the size of proteome, for ease of visualization.

The likelihood of a phylum having HPs was calculated by dividing the number of HPs by the size of the proteome, and multiplying the result by 100. The same filtering step described above was performed. For visualization, we sorted each phylum first by their corresponding superkingdom and after by the median likelihood.

### Huge protein existence

The first metric used to better understand if these are indeed existing sequences, was Uniprot Existence Level (https://www.uniprot.org/help/protein_existence). This metric visualizes the level for each protein regarding the type of evidence that supports its existence. If there is evidence at protein level, this supersedes evidence at the transcript level, even though it might exist. According to Uniprot: "Only the highest or most reliable level of supporting evidence for the existence of a protein is displayed for each entry".

To obtain the comparison between HPs and Non-HPs Existence level, first we annotated each HP with the corresponding level assignment from Uniprot. To obtain the number of Non-HPs with a specific Existence level, we retrieved the information available in the statistics page for the current Uniprot release (https://web.expasy.org/docs/relnotes/relstat.html and https://www.ebi.ac.uk/uniprot/TrEMBLstats). With this information, we calculated the corresponding percentage of each level contribution for either dataset. Besides the proportion, we also tested sampling the Non-HPs dataset, with each sample having the same size as the HPs dataset (41 754 proteins). Proteins from each sample were then annotated with the corresponding level assignment, following the same procedure as above. Lastly, Non-HPs counts

from each category were used to derive an average and standard deviation. We repeated this 5, 10 and 100 times to understand the change in the overall trend. In both cases, the "Uncertain" level was not used, as it is not present in HPs and is a very small percentage of the Non-HPs (~ 0.001%). Both the one-way ANOVA and kf were performed using the Scipy stats package.

To obtain the BUSCO Completeness Score for each of the proteomes in our dataset, we queried Uniprot Proteomes. With this information we compared each of the variables (either HP length or the number of HP per Proteome) against the corresponding Completeness Score. The Pearson correlation was calculated for each superkingdom separately.

The comparison between the Number of HPs per proteome and the assembly level was performed only for the HP dataset. To obtain this we queried Uniprot Proteomes, retrieving the Component data. Each category was obtained by parsing the corresponding Component name, which we called "Assembly Level".

## Clustering of orthologues

Clustering of HPs based on sequence similarity was performed using mmseq2 (version 13-45111) [29]. All proteins were clustered with minimal sequence identity of 0.3, coverage mode 0 and the alignment coverage of 0.8. The minimal sequence identity was chosen based on [30], that proposed a threshold for homology between proteins that is between 0.25 and 0.35.

Orthology of the clusters, and functional annotation, was obtained through eggnog-mapper (web sever version 2.1.9) [31], using the default parameters.

Manipulation and visualization of clusters was performed using Cytoscape (version 3.9.1) [32].

## Pfam domain annotation

Domain annotation of each HP was performed by hmmscan (version 3.3.2) [33] against PfamA, using 0.001 e-value threshold and 0.001 domain e-value as parameters. We then parsed the output in "–domtblout" format to fit our current dataset, using an in-house python script. To obtain the architectures(i.e. ordered combination of domains), first we structured the output such that for each protein, we mapped the pfam accessions to the number of times it occurred in the protein and to the corresponding coordinates given by hmmscan. Before plotting, all possible pfam accessions were mapped to a unique color for consistency and ease of comparison among architectures. Finally, in order to aggregate every protein with the same architecture, we first transformed each deduced architecture into an alphanumeric pattern. This pattern was used to compare the protein architectures in all-against-all procedure. Each group obtained was assigned a unique 6 character code to facilitate the investigation of distribution of HP architectures. all analyses were performed with an in-house python script.

## Non-domain centric annotations

Annotation for transmembrane helices (TMH) was performed using DeepTMHMM [34], signal peptides using SignalP – 6.0 [35] and disorder prediction using IUPred3 [36]. For TMH only the number of predicted transmembrane regions was considered. For signal peptides, only the presence or absence was considered. Finally for disorder, only the percentage was considered, either for HPs or a sampled subset of Non-HPs (n=41 754). Mapping of gene ontology terms was made by querying Uniprot for each of the proteins of interest, obtaining information for any of the categories (Molecular Function, Biological Process, Cellular Component). The most up-to-date obo file (http://purl.obolibrary.org/obo/go.obo) was used to navigate the overall ontology. A list of unique GO's for each category for the whole dataset was used to visualize the most common assignments for each superkingdom.

## Supporting information

**S1 Fig. Close-up of the distribution of all the protein lengths up to 5000 aminoacids.** Showing the considerable number after the protein length average and before the 5000 threshold.
(TIFF)

**S2 Fig. All Huge protein clusters.** All clusters found colored by phyla, (A) Archaea, (B) Bacteria and (C) Eukaryotes.
(TIFF)

**S3 Fig. Protein features and localization.** (A) Scatter plot the Disorder percentage in relation to Protein Length. On the right is the same analysis, for Non-Huge proteins (less than 5000 aa's). From top to bottom divided between Prokaryotes and Eukaryotes, respectively. Each subplot is followed by a histogram of disorder percentages. (B) Histogram of presence (red) or absence (blue) of signal peptides in relation to Protein Length. The y-axis is in logarithmic scale. (C) Density plot of the distribution of number of Transmembrane helices (TMHs) in relation to Protein Length, with corresponding rug-plot. Colored based on number of TMHs. The x-axis is in logarithmic scale.
(TIFF)

**S4 Fig. Distribution of Gene Ontology terms (GOterms), divided by superkingdom.** (A-C) corresponds to molecular function (MF), biological process (BP) and cellular component (CC), respectively. Each protein can contribute more than one GOterm to each category.
(TIFF)

**S1 Table. Most common EggNog function by superkingdom.**
(TSV)

**S2 Table. All potential unreal huge proteins.**
(TSV)

**S3 Table. Filtered potential unreal huge proteins.**
(TSV)

## Acknowledgments

We would like to thank all members of the Microplatypus group (CABD) for their helpful insights and suggestions.

## Author Contributions

**Conceptualization:** Damien P. Devos.

**Data curation:** Anibal S. Amaral.

**Formal analysis:** Anibal S. Amaral, Damien P. Devos.

**Funding acquisition:** Damien P. Devos.

**Investigation:** Anibal S. Amaral.

**Methodology:** Damien P. Devos.

**Project administration:** Damien P. Devos.

**Supervision:** Damien P. Devos.

**Validation:** Damien P. Devos.

**Visualization:** Damien P. Devos.

**Writing – original draft:** Anibal S. Amaral, Damien P. Devos.

**Writing – review & editing:** Anibal S. Amaral, Damien P. Devos.

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
