## [Decision Letter · Decision Letter 0]

11 Mar 2024

Dear Dr Devos,

Thank you very much for submitting your manuscript "The neglected giants: Uncovering the prevalence and functional groups of huge proteins in proteomes" for consideration at PLOS Computational Biology.

As with all papers reviewed by the journal, your manuscript was reviewed by members of the editorial board and by several independent reviewers. In light of the reviews (below this email), we would like to invite the resubmission of a significantly-revised version that takes into account the reviewers' comments.

While the reviews seem quite clear and helpful overall, we would like to invite you to pay particular attention to their suggestions regarding the statistical support, checking at the nt level the data not being artifacts and the clarity of the presentation, including the writing and the legibility of figures.

We cannot make any decision about publication until we have seen the revised manuscript and your response to the reviewers' comments. Your revised manuscript is also likely to be sent to reviewers for further evaluation.

Sincerely,

Erich Bornberg-Bauer

Guest Editor

PLOS Computational Biology

Arne Elofsson

Section Editor

PLOS Computational Biology

Reviewer's Responses to Questions

**Comments to the Authors:**

Reviewer #1: The manuscript "The neglected giants: Uncovering the prevalence and functional groups of huge proteins in proteomes" by Amaral and Devos, investigates the phenomenom of large proteins beyond 5000 amino acids in length. A special emphasis of their study is on the "confirmation of existence" of those huge proteins through several proxy metrics to falsify is they are artifacts of coding predictions or other methods. This is a rather small, initial study on a yet understudied topic. I think its a fairly good approach of the authors to first examine if those proteins are not artifactual.

Major:

- Please analyse how repetitive and complex the composition of these protein sequences are. This could give further indications on how they evolved.

- Figure 2A and 2B, please conduct appropriate statistical tests on this data. Also, the low values in this Figure are hard to see and paritally covered by the orange-dotted box.

- Statistical values should be stated in the text of the result sections and not simply termed "slight positive correlation" etc., if statistical tests were actually performed. This is not clear to me from reading of the manuscript and is a major issue which will have to be fixed.

- Same problem lines 102-112

- Generally, the text of the results section is very independent of the methods. Would like to have one short sequence for each result which tools are used. Or even only brackets. Now it reads like it comes all out of thin air

- line 167-182, please provide values.

Minor:

- Citation for the first claim (line 44)

- Sudden switch in text in lines 81-85. Feels deattached to the rest of the introduction.

- Why exactly is the mere existence of such proteins surprising (line 117)? Please elaborate.

- Fig 3A and 3B, numbers in the legend for proteins in those cluster would be helpful

- Line 206, please elaborate how these huge proteins could be novel ones? Novel or homology-lacking proteins were yet mostly shorter proteins or rather peptides. In my opinion, this could be an addtitional and interesting point of this study.

- Please add the phyla to your Supplementary table 2

- Some Figures are hard to read (very small font) and seem a bit overstretched e. g. Figure 1D

Reviewer #2: This is a potentially very interesting piece of work, investigating the extremes in terms of open reading frame length. At the outset I very much wanted to like this work, and I remain supportive. But I think there is a little too much optimism here and some additional due diligence to ensure that we are not just observing database issues needs to be done. In particular there is no attempt to use nucleotide data to support that the predicted ORFs are actually real (nucleotide frequencies can be very different between expressed regions and intergenic sequences for example). I also felt that the discussion does not fully expose the possible interpretations and evolutionary models - the Huntington’s example seems to me to be a bot silly when dealing with such disparate sizes. The obvious interpretation from the examples given in F4 would presumably be expansion or fusion of an array of paralogues as four of the examples are clearly repetitive. Specific comments are below;

Advances in biology often arise from the exploration of extremes.

Sweeping statement and not clear it is true.

5.000 amino acids

Odd; usually either 5 000 or 5,000? As is this means five…

majority found in Eukaryotes

Frequency per million ORFs?

Abstract and summary are very similar

Alternate between colossal, massive and huge. Suggest decide on one term throughout?

L66 Nonribosmal

L87 Where do they occur?

F1B Need to make clear this plot starts at 5k. Suggest in A use a different color to avoid confusion?

F1D Seems a missed opportunity to not include some Asgard Archaea?

F2A I think this needs better explanation as this seems a bit odd to me. The idea that less than 1% of proteins in the databases have evidence of existence at transcript level seems very low or am I misreading this. I’m also unclear what homology means in the present context - is this a single domain or across the entire length? And in this case what does predicted mean - simply what is in database but lacks other evidence? And finally bin 5 seems empty; please explain.

How accounting for erroneous intron predictions?

What is the quality cutoff for data concerning transcripts/protein detections? Obviously this is somewhat arbitrary but I think this needs to be in the main text at least in brief. Methods are rather brief.

L149 Thus, the existence of HPs can be, indirectly and for the most part, linked to higher genome quality.

Grammatically this is very confusing.

F3 What is cut-off for largest clusters? Appear to be different genomes sampled for Archaea here to F1. Is this the case? And also for panels A and B? Not sure the point of panel D. I notice that the data in B are heavily Opistokhonta biased. Is there a reason for this? My bacteria taxonomy knowledge is not good enough to know how well the sampling here is - comment?

L167 The majority of Eukaryotic clusters are related to key intracellular processes.

Pretty true for the majority of any eukaryotic protein?

In F3 and T1 I am not seeing any attempt at statistical analysis. Also, as these proteins are huge, how many domains are there that are predicted for each? Point being that as there are possibly many domains the connection with a specific functionality may be difficult if, for example, domains that participate in different pathways are predicted as present in the same ORF. I’d really like there to be a more detailed analysis of at least a few examples, where there is confident evidence for expression. Appreciate this is in part in F4, but more examples would help.

I don’t really understand the basis for the ‘Unreal’ category. This seems to be a statement of the obvious in that some of these predictions are going to be incorrect. Many gene annotations are known incorrect due to our lack of complete understanding of splicing, initiation and termination signals.

F4 The prominence of clear domain repeats is not discussed. It is of specific interest as to how common the feature is across the HPs as it suggests a possible evolutionary model.

L267 As expected, the majority belongs to Eukaryotic proteomes, followed by Bacteria and very few representatives from Archaea

But this is a result of the bias in taxon sampling? Needs to be keyed to frequency?

Reviewer #3: This study describes a comprehensive bioinformatic analysis of large proteins or “huge proteins”, those proteins exceeding 5,000 amino acids, which are often overlooked in biological research. The authors generated a set of 41,000 of these huge proteins from UniprotKB and then analyzed different aspects, including prevalence in difference organisms, likelihood to exist, clusters of orthologs, functions and functional domains, and abundance of structural features such as protein disorder and transmembrane helices.

The study makes several interesting observations. For example, huge proteins are often involved in key cellular processes, such as cytoskeleton organization and functioning as chaperones or E3-ubiquitin ligases in Eukaryotes, where in bacteria they primarily contribute to non-ribosomal peptide synthesis, polyketide synthesis, and play roles in pathogen-host interactions. The study underlines the importance of further exploring these giant proteins for their cellular roles, ecological significance, and potential biotechnological applications.

Overall, this is a well-written and interesting study addressing an unexplored topic. Below I include comments and suggestions to improve this work.

The observation in lines 195-196 is intriguing: “Around 1442 proteins have no homology inside our dataset, but are assigned to COGs, suggesting a possible recent fusion of smaller proteins into HPs.” Is there a computational method to verify this hypothesis, possibly through domain alignments?

Could the authors consider classifying those proteins not found in COGs by alternative methods? For instance, identifying common domains? Previous studies by the same group have uncovered important evolutionary relationships beyond sequence similarity. I could imagine that the standard definitions of orthologs using sequence similarity alone could potentially not capture distant biological relationships and even do this analysis on the whole set including on proteins with assigned COG.

The COG annotations listed in Table 1 are quite general. A more detailed categorization could be beneficial.

The analysis of various structural features seems brief, and I think it can add more to the study. Perhaps additional features should be included in the analysis (e.g. coiled coil regions, protein protein interactions). Further, the presence of a 30% disordered structure in a 10,000 amino acid protein warrants a deeper examination. Are these proteins characterized by more disordered domains, do they consist of numerous linkers, or do they correspond to the coiled coil type? The illustration in Figure 4 seems to indicate a combination. As recent studies reveal, disordered regions come in different forms. Due to the high abundance of disordered regions in huge proteins and their likely importance additional analysis should be conducted.

Corrections for several typos:

Line 72: change 'where' to 'were'

Line 151: change 'Fig3' to 'Fig 3'

Line 363: change 'this' to 'these'

Line 393: replace '. . ' with a single period '.'

**Have the authors made all data and (if applicable) computational code underlying the findings in their manuscript fully available?**

Reviewer #1: Yes

Reviewer #2: **No: **Not made explicit in the methods.

Reviewer #3: Yes

PLOS authors have the option to publish the peer review history of their article (what does this mean?). If published, this will include your full peer review and any attached files.

Reviewer #1: No

Reviewer #2: No

Reviewer #3: No
---

## [Decision Letter · Decision Letter 1]

8 Jul 2024

Dear Dr Devos,

Thank you very much for submitting your manuscript "The neglected giants: Uncovering the prevalence and functional groups of huge proteins in proteomes" for consideration at PLOS Computational Biology. As with all papers reviewed by the journal, your manuscript was reviewed by members of the editorial board and by several independent reviewers. The reviewers appreciated the attention to an important topic. Based on the reviews, we are likely to accept this manuscript for publication, providing that you modify the manuscript according to the review recommendations.

Please adress comments from Reviwer 2 before submitting the final version.

Sincerely,

Arne Elofsson

Section Editor

PLOS Computational Biology

Arne Elofsson

Section Editor

PLOS Computational Biology

Please adress comments from Reviwer 2 before submitting the final version.

Reviewer's Responses to Questions

**Comments to the Authors:**

Reviewer #1: The authors have addressed my concerns.

Reviewer #2: The majority of proteins have a length between 100 and 500 aminoacids (aa), approximately [1]..

Not clear how a ramge is approximate.

presence of numerous proteins exceeding 5000 amino acids in length

Previous sentence has aa as abreviatio for amino acids, so use it.

still Huge genes and huge genes

if using an abbrev then use it….

Drosophila Melanogaster. ooops

_

_has hardly been investigated

better - not thoroughjly

where the organism is not growing

better - not proliferating

This protein, with almost 10 000 aa, has been found to bind Fibronectin

fibronectin

and the recent descriptions of Omnitrophota species which have been found to

possess many HP some of which are the largest ever reported, even larger than eukaryotic one

and recent descriptions of Omnitrophota species which possess many HP, including some of the largest ever reported, exceeding the size of HPs suggested for eukaryotes.

However, recent broad analyses are missing.

So????? What to address?

we decided to answer a broad range of questions

address better

I have not put in additional issues in the wroting and hope that the authors will edit the MS carefully.

**Have the authors made all data and (if applicable) computational code underlying the findings in their manuscript fully available?**

Reviewer #1: Yes

Reviewer #2: Yes

PLOS authors have the option to publish the peer review history of their article (what does this mean?). If published, this will include your full peer review and any attached files.

Reviewer #1: **Yes: **Lars Eicholt

Reviewer #2: No

Figure Files:

Data Requirements:

Reproducibility:

References:

---

## [Decision Letter · Decision Letter 2]

4 Sep 2024

Dear Dr Devos,

We are pleased to inform you that your manuscript 'The neglected giants: Uncovering the prevalence and functional groups of huge proteins in proteomes' has been provisionally accepted for publication in PLOS Computational Biology.

Best regards,

Erich Bornberg-Bauer

Guest Editor

PLOS Computational Biology

Arne Elofsson

Section Editor

PLOS Computational Biology

Reviewer's Responses to Questions

**Comments to the Authors:**

Reviewer #2: My apologies for asking once more for a language/grammer edit, but I found many instances of poor english that I really thing the uthors would want to address. A few examples, bu tthere are many;

evidences should be evidence

If anything, a slight positive correlation between genome completeness and number of HPs can be appreciated.

Change appreciated to detected, observed or similar

a

Similarly, no correlation between the genome assembly levels

Either: Similarly, no correlation between genome assembly levels… or… Similarly, no correlation

between the genome assembly level

Is Bacteria upper case or not? Please check as I tend to think is ‘bacteria’. Similarly eukaryotes and not Eukaryotes, and the names of most domians are not normally formalised, so should be lower case.

Finally, for Archaea, the most common domain, by a large margin,

‘large margin’ is really a slang term, aka ‘whopper’. Suggest ‘significantly’

**Have the authors made all data and (if applicable) computational code underlying the findings in their manuscript fully available?**

Reviewer #2: Yes

PLOS authors have the option to publish the peer review history of their article (what does this mean?). If published, this will include your full peer review and any attached files.

Reviewer #2: No

---

## [Editor Report · Acceptance letter]

9 Sep 2024

PCOMPBIOL-D-23-01891R2 

The neglected giants: Uncovering the prevalence and functional groups of huge proteins in proteomes

Dear Dr Devos,

I am pleased to inform you that your manuscript has been formally accepted for publication in PLOS Computational Biology. Your manuscript is now with our production department and you will be notified of the publication date in due course.

With kind regards,

Zsofia Freund
